# On Automated Object Grasping for Intelligent Prosthetic Hands Using Machine Learning

**DOI:** 10.3390/bioengineering11020108

**Published:** 2024-01-24

**Authors:** Jethro Odeyemi, Akinola Ogbeyemi, Kelvin Wong, Wenjun Zhang

**Affiliations:** Advanced Engineering Design Laboratory, Division of Biomedical Engineering, University of Saskatchewan, Saskatoon, SK S7N 5A9, Canada; rye164@usask.ca (J.O.); akinola.ogbeyemi@usask.ca (A.O.); kelvin.wong@usask.ca (K.W.)

**Keywords:** computer vision, electromyography, hand gestures, machine learning, prosthetics

## Abstract

Prosthetic technology has witnessed remarkable advancements, yet challenges persist in achieving autonomous grasping control while ensuring the user’s experience is not compromised. Current electronic prosthetics often require extensive training for users to gain fine motor control over the prosthetic fingers, hindering their usability and acceptance. To address this challenge and improve the autonomy of prosthetics, this paper proposes an automated method that leverages computer vision-based techniques and machine learning algorithms. In this study, three reinforcement learning algorithms, namely Soft Actor-Critic (SAC), Deep Q-Network (DQN), and Proximal Policy Optimization (PPO), are employed to train agents for automated grasping tasks. The results indicate that the SAC algorithm achieves the highest success rate of 99% among the three algorithms at just under 200,000 timesteps. This research also shows that an object’s physical characteristics can affect the agent’s ability to learn an optimal policy. Moreover, the findings highlight the potential of the SAC algorithm in developing intelligent prosthetic hands with automatic object-gripping capabilities.

## 1. Introduction

Prosthetic organs, e.g., hands [1,2], are a robotic system that has a power generator, actuator, body, sensor, and controller [3,4]. One of the important challenges with prosthetic organs, e.g., hands, is to control the prosthetic hands from the patient’s brain. Mapping human intentions to the prosthetic hand’s actuator when performing a task is a challenge. A popular approach is to make use of electromyography (EMG) signals to represent the patient’s intent to perform a task. Therefore, EMG-controlled prostheses are popular today, which incorporate embedded motors that move the fingers based on recorded electrical activities during muscle contractions that come from the patient’s brain [5]. These prostheses allow for multiple grip patterns and user-friendly interaction [6]. However, a major challenge with EMG prosthetics is that it takes users many months of continuous training to achieve full mastery over EMG-controlled prosthetics [7]. The training process takes place through repetitive exercises where the user learns to consciously control their muscle contraction. In practice, because of this strenuous training process, many users abandon the prosthesis before mastery is achieved. Another challenge with the EMG-controlled prosthetic hand is the inherent difficulty in representing high-resolution brain signals for task manipulation; often, in practice, the failure to grasp an object often happens, either with the object slipping out of the prosthetic hand or finger or the object being damaged due to too the high force from the prosthetic hand [8]. Several methods to enhance the EMG representation have been proposed, including electroencephalography (EEG) [9], eye tracking, transfer learning [10,11], and facial recognition. EMG, along with its enhancing methods, is limited because the desired resolution with the patient brain signal (e.g., to grasp an object) increases far faster than the resolution of the representation of patient intent in the brain that can be achieved with the current technology such as EMG, EEG, and so on.

The other idea to overcome the limit is to make the prosthetic hand highly autonomous, meaning that it can automatically perform the task of gripping an object. Hao et al. (2021) designed a low-cost soft prosthetic hand with embedded actuation and sensors, allowing for initial contact detection during gripping to prevent damage to objects [12]. However, the inclusion of extra sensors increased the weight of the prosthetic hand and posed the risk of sensor degradation. Castro et al. (2022) presented a prototype incorporating EMG-based control and computer vision to control a prosthetic hand seamlessly with different grip patterns [13]. However, their paper did not address the control of grip pressure, which is crucial when handling delicate objects. Czimmermann et al. (2020) conducted a comprehensive review of defect detection technologies, emphasizing the resource requirements for effective training of neural networks [14]. They also highlighted the challenges of parallelization when dealing with large datasets. Abbasi et al. (2019) used unsupervised learning techniques to analyze patterns in different grasp types, but their approach was deemed expensive and impractical [15].

The motivation of the study presented in this paper was to advance the technology along the idea, i.e., improving the autonomy of the prosthetic hand. Current prosthetics lack the ability to automatically grip objects or perform hand gestures without extensive user training or adding excessive weight. This study addresses this gap by exploring machine learning techniques to develop prosthetic arms capable of automatic grip control and hand gestures, offering a potential breakthrough in prosthetic technology for enhanced usability and efficiency. The objective of this paper is to train an agent (i.e., prosthetic hand) using three different algorithms—Soft Actor-Critic (SAC) [16], Proximal Policy Optimization (PPO) [17], and Deep Q Networks (DQN)—to enable the prosthetic hand to autonomously grasp objects. To achieve the objective, the impact of each algorithm on the effectiveness of the gripping action will be evaluated [18,19], and an optimal model for a variety of objects will be tested, specifically investigating how the physical properties of the objects may significantly influence the capability of the prosthetic hand to grip them effectively. 

This paper presents significant contributions to the field of Myo prosthetic hand development by leveraging computer vision and machine learning technologies. Also, this paper will provide an understanding of the impact of object properties on grasping success. These contributions are important in improving the development of more efficient and effective prosthetic grasping systems with low cost and high usability.

## 2. Background

### 2.1. Reinforcement Learning

Reinforcement learning (RL) forms a major part of this study. The prosthetic hand end-effector learns to grip objects of different shapes with the right amount of force in various environments using different reward models and policies. Reinforcement learning is a subfield of machine learning that deals with how intelligent agents can learn to make decisions in dynamic environments by taking actions that maximize a cumulative reward [20]. It is based on the idea of trial-and-error learning, where the agent learns from its experiences with the environment rather than relying on pre-programmed rules. Rewards and penalties are used in reinforcement learning to communicate appropriate and inappropriate behavior.

The goal of reinforcement learning is to form algorithms that can learn how to make good decisions in complex, uncertain, and dynamic environments. This makes it well-suited for a wide range of applications, such as game playing, robotics, autonomous driving, and resource management, among others. At the heart of reinforcement learning is the concept of an agent that interacts with an environment. The environment is typically modelled as a Markov decision process (MDP), which is a mathematical framework used to model decision-making in situations where outcomes are uncertain and influenced by both random chances and agent actions. A Markov decision process (Figure 1) is formally defined as a tuple (S, A, T, R, and γ), where S is the state space, A is the action space, T is the transition function, R is the reward function, and γ is the discount factor. An MDP is a stochastic process, where the state of the environment evolves over time according to the transition function T. At each time step, the agent observes the current state (s), selects an action a from the action space A, and receives a reward r from the reward function R. The goal of the agent is to learn a policy π: S → A that maps each state to an action such that the expected cumulative reward is maximized.

In an MDP, the probability that an agent can get to a state (s’) from a state s at time t by action a can be represented by Equation (1):(1)$$P_{\propto }\left(s,s^{\prime }\right)=P_{r}\left(s_{t+1}=s^{\prime }\right|s_{t}=s,a_{t}=a).$$

If the agent were to receive a reward for each transition between states, Equation (1) can be extended to Equation (2):(2)$$P_{\propto }\left(r|s,s^{\prime }\right)=P_{r}\left(s_{t+1}=s^{\prime },R_{t}=r\right|s_{t}=s,a_{t}=a).$$

### 2.2. Soft Actor-Critic

The soft actor-critic (SAC) algorithm (Algorithm 1) is a practical approximation to soft policy iteration that uses function approximators for both the Q-function and the policy [21]. It alternates between optimizing both networks with a stochastic gradient descent. The parameters of these networks are ψ, θ, and φ. The state value function approximates the soft value, which can be estimated from a single action sample from the current policy. However, including a separate function approximator for the soft value can stabilize training and make it convenient to train simultaneously with the other networks. The soft value function is trained to minimize the squared residual error (Equation (3)):(3)$$J_{V}\left(\varphi \right)=\in_{s_{t}}\sim D\left[\frac{1}{2}{\left(V_{\varphi }\left(S_{t}\right)-{\in_{a}}_{t\sim \pi_{\varphi }}\left[Q_{\theta }\left(S_{t},a_{t}\right)-log\pi_{\emptyset \left(a_{t}|S_{t}\right)}\right]\right)}^{2}\right].$$
**Algorithm 1.** Soft Actor-Critic [21]. 
$\mathrm{Initialize}\;\mathrm{parameter}\;\mathrm{vectors}\;\psi ,\bar{\psi },\theta ,\phi$.
**  for** each iteration **do**
**     for** each iteration step **do**
$\;\;\;\;\;a_{\left(t\right)}∼\pi_{\left(\phi \right)}\left(a_{\left(t\right)}\left|\left(s_{\left(t\right)}\right)\right|\right)$
$\;\;\;\;\;s_{\left(t+1\right)}∼p\left(s_{\left(t+1\right)}|s_{\left(t\right)},a_{\left(t\right)}\right)$
$\;\;\;\;\;D\leftarrow D\cup \left(s_{\left(t\right)},a_{\left(t\right)},r\left(s_{\left(t\right)},a_{\left(t\right)}\right),s_{\left(t+1\right)}\right)$
**    end for**
**    for each gradient step do**
$\;\;\;\;\;\psi \leftarrow \psi -\lambda_{\left(V\right)}\nabla_{v}J_{\left(V\right)}\left(\psi \right)$
$\;\;\theta_{\left(i\right)}\leftarrow \theta_{\left(i\right)}-\lambda_{\left(Q\right)}\nabla_{\left(\theta_{\left(i\right)}\right)}J_{\left(Q\right)}\left(\theta_{\left(i\right)}\right)\;for\;i\in \left\{1,2\right\}$
$\;\;\;\;\;\phi larr\phi -\lambda_{\left(\pi \right)}\nabla_{\left(\phi \right)}J_{\left(\pi \right)}\left(\phi \right)$
$\;\;\;\;\;\;\;\bar{\psi }\leftarrow \tau \psi +\left(1-\tau \right)\bar{\psi }$

**    end for**
**   end for**


### 2.3. Proximal Policy Optimization

The goal of PPO (Algorithm 2) is to find a policy (a set of actions to take in different states) that maximizes the expected reward. PPO is based on the idea of a surrogate loss function that is used instead of the policy gradient, and it uses a value function to estimate the advantage of taking an action in each state [22]. To implement PPO, one first constructs a surrogate loss function that can be differentiated and optimized using stochastic gradient ascent. This surrogate loss function includes a clipping term or a KL penalty term, depending on the specific implementation. Additionally, a value function is used to estimate the advantage of taking an action in each state. The policy and value function can share parameters, which helps to reduce the number of parameters that need to be learned.
**Algorithm 2.** PPO, Actor-Critic style [22].
 **for** each iteration **do**
   **for** each actor **do**
$\;\;\;\;\;\mathrm{Run}\;\mathrm{policy}\;\pi \theta_{old}$ in environment for T timessteps
$\;\;\;\;\;\mathrm{Compute}\;\mathrm{advantage}\;\mathrm{estimates}\;A_{1}\ldots \ldots A_{T}$
    **end for**
    Optimize surrogate L wrt θ
, with K epochs and minibatch size M≤NT
$\;\;\;\;\;\Theta_{old}\leftarrow \theta$
 **end for**


### 2.4. Deep Q-Networks

Deep Q-Networks (DQN) is a variant of Q-learning that uses deep neural networks to represent the action-value function. DQN was first introduced by Mnih (2015) and has since been extended and improved in many ways [23]. DQN works by using a neural network (Algorithm 3) to approximate the action-value function. The algorithm uses a technique called experience replay to store and randomly sample from a replay buffer of past experiences, which helps to stabilize the learning process and reduce the effects of correlations between consecutive samples.
**Algorithm 3.** Deep Q-learning with experience replay [24].
Initialize replay memory D to capacity N
Initialize action-value function Q with random weights θ
Initialize target action-value function $\hat{Q}$ with weights $\theta^{-}=\theta$

**for** episode 1, M do Initialize sequence $s_{1}=\left\{x_{1}\right\}$ and preprocessed sequence $\phi_{1}=\phi \left(s_{1}\right)$
  **for** t = 1, T **do**
    With probability ε
$\;\mathrm{select}\;a\;\mathrm{random}\;\mathrm{action}\;a_{t}$
$\;\;\;\;\mathrm{otherwise}\;\mathrm{select}\;a_{t}=argmax_{\left(a\right)Q\left(\phi \left(s_{t}\right),a;\theta \right)}$
$\;\;\;\;\mathrm{Execute}\;\mathrm{action}\;a\text{\_}t\;\mathrm{in}\;\mathrm{the}\;\mathrm{emulator}\;\mathrm{and}\;\mathrm{observe}\;\mathrm{reward}\;r_{t}$
$\;\mathrm{and}\;\mathrm{image}\;x_{t+1}$
$\;\;\;\;\mathrm{Set}\;s_{t+1}=s_{t},a_{t},x_{t+1}$
$\;\mathrm{and}\;\mathrm{preprocess}\;\phi_{t+1}=\phi \left(s_{t+1}\right)$
$\;\;\;\;\mathrm{Store}\;\mathrm{experience}\;\left(\phi_{t},a_{t},r_{t},\phi_{t+1}\right)$ in D
**    Sample random minibatch of experiences** $\left(\phi_{j},a_{j},r_{j},\phi_{j+1}\right)$ **from D**

$\;\;\;\;\mathrm{Set}\;y_{j}=\left\{r_{j}+\begin{array}{cc} r_{j} & if\;episode\;terminates\;at\;step\;j+1\\ {\gamma max}_{\left(a^{\prime }\right)hat\left(Q\right)\left(\phi_{j+1},a^{\prime };\theta^{-}\right)} & otherwise \end{array}\right.$
    **Perform a gradient descent step on** ${\left(y_{j}-Q\left(\phi_{j},a_{j};\theta \right)\right)}^{2}$ **with respect to the weights θ**
    **Every C steps reset** $\hat{Q}=Q$
 **end for**
**end for**


## 3. Experiment

This section covers the experimental setup for using reinforcement learning to train the prosthetic arm gripper to be able to grip an object in minimum time and with optimal force. 

### 3.1. Experiment Process

The gripper was simulated and visualized in a PyBullet [25] environment, which includes a plane to set the world and a table at the center (Figure 2). PyBullet is a physics simulation engine designed for use in robotics, machine learning, and computer graphics applications. For this research, we initially compared the performance of the three most popular simulation engines–Mujoco, Gazebo, and PyBullet. Mujoco is a proprietary physics engine that is particularly well-suited for simulating articulated bodies, such as humanoids and robotic arms. It provides high-fidelity simulations that are accurate and computationally efficient, making it a popular choice for research in fields such as biomechanics and robotics. However, its proprietary nature means that it can be expensive to use and limits the ability of researchers to modify or extend the engine to meet their specific needs.

Gazebo [26], on the other hand, is an open-source simulation engine that is particularly well-suited for simulating robots in dynamic environments. It provides a range of sensors and actuators, as well as tools for visualizing and analyzing simulation data, making it a popular choice for testing and validating control algorithms and overall system designs. Additionally, because it is open source, it is highly customizable and can be modified and extended to meet specific research needs. However, because it is designed to simulate complex environments, Gazebo can be computationally intensive, making it less suitable for certain types of simulations. For this reason, we chose the PyBullet simulation engine.

In the environment, objects are randomly spawned on the table from a defined height with the goal of having the gripper locate these objects and grip them. In a real-world scenario, the gripper would be mounted as the end effector of a prosthetic arm. The human would need to place the end effector over an object to be grasped, while the gripper will automatically control the gripping action by the end effector.

An RGBD camera is mounted on the gripper, which returns information on the observed state. The camera returns information to the gripper from three perspectives: RGB data, depth data, and the segmented mask.

The gripper positions itself based on actions it receives in the form of a collection of values (dx, dy, dz, ∅, and open/close gripper). These values are in the range of −1 to +1 where dx, dy, and dz represent the step translation the gripper needs to make in the x, y, and z direction, respectively. ∅ represents the yaw rotation and the last value signifies whether to open or close the gripper. Once again, these actions are in play here to run the simulation. In a real-world scenario, the gripper would need information on just the yaw and gripper open/close action since the human would have to translate the gripper in the x, y, or z directions.

Rewards or punishments are given in every episode depending on different factors, like whether the gripper accomplished its task or how long time it took to accomplish that task. An episode ends when the gripper successfully picks up an object or if it is unable to after a set number of steps. A shaped reward function is used to tailor to the specific gripping task as opposed to using a regular binary reward function.

#### Environment Assumptions

(1)The simulation environment is deterministic, i.e., given the same initial state and actions, the environment will always produce the same outcome.(2)The action space and observation space of the environment are well-defined and consistent throughout the training and testing phases.(3)No external condition can cause the objects to move other than the gripper or other spawned objects in the environment.

The environment parameters are summarized in Table 1.

### 3.2. Sensor Setup

The gripper used for this simulation is the WSG50 gripper (Figure 3). The WSG50 is a high-precision, two-finger parallel gripper designed for a wide range of automation applications.

The WSG50 gripper is provided as a built-in robot in the PyBullet library files, making it out-of-the-box compatible with the PyBullet simulation environment. The gripper is made up of 8 links and 8 joints. A two-finger gripper was used to make it easier to set up the simulation. 

### 3.3. Sensor Setup

One of the critical factors that determine the success of reinforcement learning for grasping and manipulation tasks is the quality of the training data. The training data must be diverse enough to cover the variability in object shapes, sizes, and textures that the gripper is likely to encounter in the household environment. Baris (2020) made use of random wooden blocks for the training and testing of the RL models, which do not fully represent the use case of the gripper [27]. For our experiment, we made use of the Pybullet URDF models library [28], which contains a wide variety of household objects, including cups, plates, bowls, bottles, cans, and other everyday items (Figure 4). The models were obtained from various sources, including the YCB dataset, which is a collection of 3D models of household objects.

The models in the Pybullet URDF models library have been carefully designed to accurately represent the physical properties of the real-world objects they are based on. This means that when they are used in robot grasping simulations, the results will be realistic and representative of what would happen if the robot were interacting with a real-world object.

### 3.4. Sensor Setup

The sensor in the task represents a camera mounted at the midpoint of the gripper’s base, which records observations captured in discrete steps. The observations captured by the sensor represent the state space in the Markov decision process. In an approach like [27], we implemented a perception pipeline using RGBD observations captured by the camera (Figure 5).

RGBD stands for red, green, blue, and depth, and refers to a type of imaging technology that combines traditional RGB color data with depth information. The method to preprocess the RGBD data before feeding it into the convolutional neural network is like the method implemented in [27]. Since our task involves a gripper with a certain width, we need to include this information in our processing of the sensor output. To achieve this, we pad the gripper width information into a three-dimensional array that has the same shape as the sensor data (i.e., 64 × 64 × 1 or 64 × 64 × 4). This extra information is added as an extra “layer” to the sensor data, which we call a “channel”.

By doing this, we can make sure that the width information of the actuator is included in the processing of the sensor data without changing the way we process it. During training, we removed this extra channel so that the robot could learn how to use the sensor data to control its movements, considering the width of the gripper. By including the gripper width information in this way, we can ensure that the robot can use this information to perform the task as we want it to do.

The RGBD data is then fed into a convolutional neural network to learn meaningful features and representations of the scene, which is used as input to the agent’s decision-making process. The output is passed through one or more fully connected layers, which are also sometimes called dense layers. The output of the last fully connected layer can then be flattened into a one-dimensional array. The difference in the CNN architecture from the work of [29] and the one used in this study is the introduction of dropout layers [30], which determine the fraction of the input units that drop out during the training of a neural network. The importance of this is to avoid overfitting, which is a recurring problem in machine learning. We also employed an exponential linear unit because of its success in speeding up learning compared to other LUs and in dealing with sparse data. After the tensor output has been flattened into a one-dimensional array, the unpadded gripper width information is then concatenated to the RGBD tensor array to form the observation vector. This process is represented in the schematic shown in Figure 6.

### 3.5. Reward Function

The reward function [31] implemented for this task is a custom-shaped reward function designed to incentivize the gripper’s behavior in the environment. The aim of the reward function is to incentivize the agent to grasp the object efficiently and to lift it to the desired height quickly while penalizing the agent for taking too long to complete the task. The reward function consists of several parameters that influence the reward value and is summarized in Table 2. Whether an object has been grasped is detected by checking the gripper width after the robot runs a close gripper event. The reward function parameters used are like [27]; however, there were a few problems with the implementation of the reward function in [27], which we attempted to solve.

(1)Lack of exploration: the reward function does not provide any incentives for the agent to explore the environment. This can lead to the agent getting stuck in local optima and failing to discover better solutions. To solve this, we introduced a random exploration component to the reward signal. This is important because it encourages the agent to take actions that it has not taken before, which can help it discover new ways to interact with the environment and achieve the task.(2)Sparse rewards: the reward function is designed to provide a large reward only when the arm lifts the object by the desired amount. This can make learning difficult for the agent, as it may take a long time to receive any useful feedback. By adding a distance penalty, we encourage the agent to move the object towards the target position. This helps to alleviate the issue of sparse rewards, as the agent can receive a small reward for making progress towards the goal, even if it has not fully achieved it yet.

### 3.6. Training Process

The agent was trained using three existing policies: SAC, DQN, and PPO. The choice of these algorithms was based on their proven effectiveness and wide usage in the field [29]. By leveraging the strengths of SAC, DQN, and PPO, it was anticipated that the agent could benefit from their advantages in terms of stability, sample efficiency, and performance.

Training begins at the reset state where the objects have spawned on the table, and the gripper also spawned midair. For each of the algorithms, we defined the reward function that provides positive feedback when the agent takes actions to move it closer to the goal and negative feedback when it takes actions to move it further away from the goal. Next, the agent is run through a series of episodes in timesteps of 1000 in the environment. Each episode consists of the agent acting and receiving rewards. The agent’s goal is to maximize the cumulative reward over all episodes. During each episode, the agent observes the current state of the environment, chooses an action based on its current policy and receives a reward based on the outcome of the action. The agent then updates its policy based on the observed rewards and the current state of the environment.

The process of updating the policy varies depending on which algorithm is being tried, but it follows a similar update rule (4). During training, the agent explores the environment to learn the optimal policy. The training process continues until the agent has the maximum number of episodes, which is predefined:(4)$$Q\left(S_{t},A_{t}\right)\leftarrow Q\left(S_{t},A_{t}\right)+\alpha [R_{t+1}+\gamma \underset{a}{\underbrace{max}}Q\left(S_{t+1},a\right)-Q\left(S_{t},A_{t}\right)].$$

$Q\left(S_{t},A_{t}\right)$ is the estimate of the action-value function at time t, $R_{t+1}$ is the reward received at time t, ∝ is the learning rate, γ is the discount factor, and $\underset{a}{\underbrace{max}}Q\left(S_{t+1},a\right)$ is the maximum value over all actions that can be taken in the next state.

## 4. Results and Discussions

### 4.1. Algorithm-Specific Success Rate

Three different algorithms were tested during training: DQN, SAC, and PPO. The training went on for 1,000,000 timesteps, and the performance of each algorithm by success rate is shown in Figure 7.

From the results above, we can see that SAC outperformed the other two algorithms in terms of success rate by converging the quickest in just under 200,000 timesteps. On the DQN algorithm, the agent never fully learns the optimal policy required for the grasping task. We can see it reaches a peak success rate of 80% in about 450,000 timesteps and then stays at equilibrium at that point. There are several possible reasons why SAC performed better than the other two algorithms. First, SAC is an off-policy actor-critic algorithm that uses a soft value function to estimate the Q-value of actions. This soft value function provides a smoother estimate of the Q-value than the hard value function used in DQN and PPO, which can lead to better performance in the action spaces. SAC also uses a stochastic policy that is updated using a combination of maximum entropy reinforcement learning and entropy regularization. This encourages the policy to explore the state space more thoroughly, which can lead to better performance in environments with sparse rewards, such as the object-picking task in our study. DQN might have performed badly because it is an on-policy algorithm that suffers from instability when learning from action spaces.

From the results, the SAC algorithm performed the best, with a mean success rate of 0.990. This is significantly higher than the mean success rate of the DQN algorithm (0.602) and the PPO algorithm (0.821). The high success rate achieved by the SAC algorithm can be attributed to its inherent advantages in handling high-dimensional action spaces. 

On the other hand, the DQN algorithm performed the worst, with a mean success rate of 0.602. The lower success rate achieved by the DQN algorithm can be attributed to the fact that the Q-learning architecture employed by the DQN algorithm is known to have stability issues when applied to continuous control tasks, as it tends to overestimate the Q-values and leads to suboptimal policies. This result is summarized in Table 3.

The MANOVA test was further conducted on the success rates of the three algorithms, DQN, SAC, and PPO, to investigate whether there were significant differences in their performance. The results are shown in Table 4. Based on the results of the MANOVA analysis, there is strong evidence to suggest that there are significant differences between the three different algorithms in terms of their performance. The Pillai’s trace statistic of 0.57982 and the highly significant *p*-value (<2.2 × 10^−16^) indicate that the different algorithms have a substantial impact on the success rate of the gripper. The approximate F-value of 2040.7 further supports the presence of a significant effect. These findings indicate that the choice of algorithm significantly influences the observed differences in the performance of the prosthetic gripper.

The Tukey HSD post hoc analysis (Table 5) was conducted to delve deeper into the observed differences in mean success rates among the three algorithms: DQN, SAC, and PPO. Specifically, the comparison between DQN and SAC revealed a mean success rate difference of 0.1689, with an adjusted *p*-value of 0.0. This underscores the considerable dissimilarity in success rates between DQN and SAC, emphasizing the superior performance of SAC in the context of object grasping. Furthermore, in the comparison between PPO and SAC, a substantial mean success rate difference of 0.3882 was observed, with an adjusted *p*-value of 0.0. This result reinforces the conclusion that SAC outperforms PPO significantly in terms of mean success rates. The null hypothesis is also rejected, emphasizing the robustness of SAC compared to PPO, making it the most effective algorithm in the object-grasping experiment.

To provide a more comprehensive understanding of the comparative performance of the algorithms, we conducted a Friedman rank test. This non-parametric test evaluates whether there are significant differences in the performance of algorithms when considering multiple related samples. The results of the Friedman test are shown in Table 6. The obtained *p*-value indicates a highly significant difference among the algorithms, consistent with our MANOVA findings. The Q-value of 29,689.74 emphasizes the substantial impact of algorithm choice on performance.

### 4.2. Comparative Analysis of SAC, PPO, and DQN for Object-Grasping–Hyperparameter Exploration

Increasing the batch size to 128 had a positive effect on the convergence speed of the SAC algorithm, enabling it to converge quickly after just 100,000 timesteps (Figure 8). This performs much better than the SAC with RGBD presented in [27].

With a larger batch size, the SAC algorithm benefits from an increased sample efficiency. More data is available for each update step, allowing for better estimating the policy gradient and reducing the impact of noise or outlier experiences. This increased data diversity enhances the stability of the learning process and facilitates faster convergence toward an optimal policy. Increasing the batch size increases the amount of information available for each update, and this significantly increases the agent’s training time.

The performance of the DQN algorithm drops at both the batch size of 32 and 128, compared to the initial analysis at batch size 64 (Figure 7). We can infer that at batch size 64, there is an optimal balance between sample efficiency and computational overhead. Deviations from this batch size, either lower or higher, lead to diminished performance due to limitations in data availability or computational inefficiencies.

We observe a similar pattern, as shown in Figure 9, when varying the number of hidden layers for the SAC and PPO algorithms. The SAC algorithms benefit from an increase in the number of hidden layers, while the PPO algorithm performs poorly in both cases at 500,000 timesteps and would possibly take a longer time to converge. Based on these results, we can make an inferred comparison of these three algorithms, as shown in Table 7.

### 4.3. Object-Specific Success Rate

Using SAC as the preferred algorithm, the agent was once again trained to grasp three different types of objects–a remote controller, soap bar, and mug (Figure 10). This is to determine how the object shape and size affect the agent convergence at an optimal policy.

The results (Figure 11) show that the mug was the easiest object for the agent to grasp, as it converged the fastest, achieving a success rate of over 0.9 after 200,000 timesteps. The remote controller was the second easiest object to grasp, with a success rate of over 0.8 after 250,000 timesteps. However, the soap bar was the most challenging object for the agent, as it failed to converge, with a success rate that remained between 0.0 and 0.008 throughout the training process.

These results are consistent with our understanding of the properties of these objects. The remote controller and mug are both easy to grip, with well-defined shapes and surfaces that the agent can easily detect and grasp. On the other hand, the soap bar has a smooth surface and is difficult to grip, making it challenging for the agent to learn a successful grasping policy.

These results can also be explained in terms of the exploration-exploitation trade-off. The agent’s task is to maximize its reward, which in this case is the success rate of grasping an object. To do this, the agent must explore different grasping strategies to find the one that maximizes its reward. However, if the exploration process is too extensive, the agent may fail to converge to an optimal policy. On the other hand, if the agent exploits the same grasping strategy without exploring others, it may miss out on better strategies.

In the case of the soap bar, the slippery surface and difficult-to-grip nature of the object may have made it more challenging for the agent to explore different grasping strategies. As a result, the agent may have become stuck in a suboptimal policy that did not lead to successful grasping. 

In contrast, the well-defined edges and surface of the remote controller and mug may have made it easier for the agent to explore different grasping strategies, leading to faster convergence to an optimal policy. Also, the fact that the success rate of these objects continued to improve over time suggests that the agent was able to find better grasping strategies through exploration without getting stuck in a suboptimal policy.

MANOVA test was further conducted on the success rates of the three objects, a mug, a remote controller, and a bar of soap, to investigate whether the type of object significantly affects the performance of the SAC algorithm. The results are shown in Table 8.

Based on the results of the MANOVA analysis, there is strong evidence to suggest that the type of object (a remote controller, soap, a mug) has a significant effect on the success rates of the prosthetic gripper. The Pillai’s trace statistic of 0.6888 and the highly significant *p*-value (<2.2 × 10^−16^) indicate that the different object types contribute significantly to the variation in the success rate of the agent. The approximate F-value of 552.9 further supports the presence of a substantial effect. These findings suggest that different objects cause the RL agent to exhibit distinct performance levels in task completion.

## 5. Conclusions

This study demonstrates the superiority of the SAC algorithm over DQN and PPO for training prosthetic hands to grasp objects effectively in terms of force and contact point. Importantly, the physical properties of the object, including shape and texture, significantly influence the success of prosthetic grasping. Recognizing these elements offers insights into creating robust prosthetic systems that prevent slippage or damage.

The SAC algorithm, with its off-policy actor-critic architecture and soft value function, outperformed other algorithms by achieving a mean success rate of 0.990. This success can be attributed to its inherent advantages in handling high-dimensional action spaces and its stochastic policy that encourages thorough exploration in environments with sparse rewards. Conversely, the DQN algorithm’s instability in continuous control tasks led to a lower mean success rate of 0.602. This can lead to the development of more intelligent prosthetics that are user-friendly and do not require extensive training with the patients for their use.

This research contributes significantly to the domain of Myo prosthetic hand development and emphasizes the advancing role of computer vision and machine learning technologies in this field. Notably, it highlights the influence of an object’s physical properties, such as shape and texture, on prosthetic grasping. The study also refines traditional autoencoder models like those presented by Breyer et al. (2019), introducing an innovative approach to sparse data handling in EMG-based applications by optimizing the learning rate for each parameter.

The methodology employed in this research can also be applied in various domains beyond Myo prosthetics. The success of the SAC algorithm suggests promising applications in industrial robotics for robot arms, automated assembly lines, warehousing, logistics, humanoid robotics, and medical robotics. The adaptability demonstrated by SAC makes it particularly suitable for tasks involving complex object manipulation.

Future work, first, the Myo armband’s second channel has hinted at significant muscle activities, warranting a deeper investigation. Understanding the distinct role of these muscles, detected by this specific channel, can unveil crucial insights. By identifying the prominence of this channel’s captured activities, there is potential to refine prosthetic algorithms, enhancing grip accuracy and responsiveness based on the underlying muscular dynamics. Second, a post hoc analysis, potentially employing methods like pairwise comparisons or interaction effect reviews, can be performed to shed light on how different algorithms perform across various objects. Considering that objects can differ in attributes such as shape, texture, and weight, it is essential to pinpoint which algorithms work best with each attribute. Understanding this can help us choose the right algorithm for specific tasks, making prosthetics work better in different real-life situations. Third, the robustness and resilience of the method will be studied. While robustness may be well known, resilience refers to whether a method still works if underlying conditions are changed (see the definition of resilience in the literature) [32,33]. 

## Figures and Tables

**Figure 1 bioengineering-11-00108-f001:**
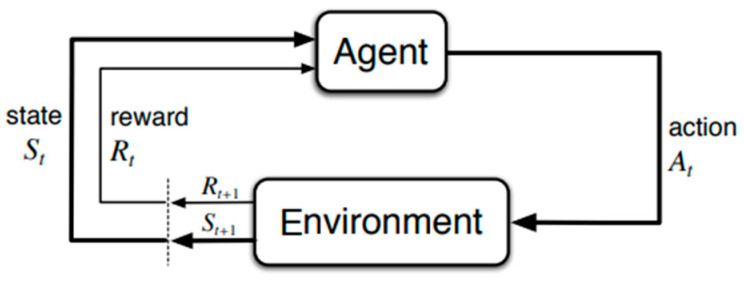
MDP Structure.

**Figure 2 bioengineering-11-00108-f002:**
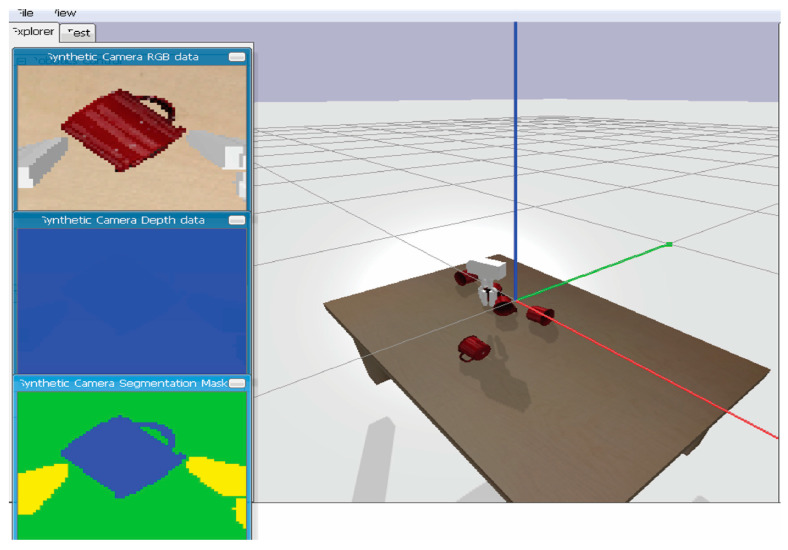
Simulation scene.

**Figure 3 bioengineering-11-00108-f003:**
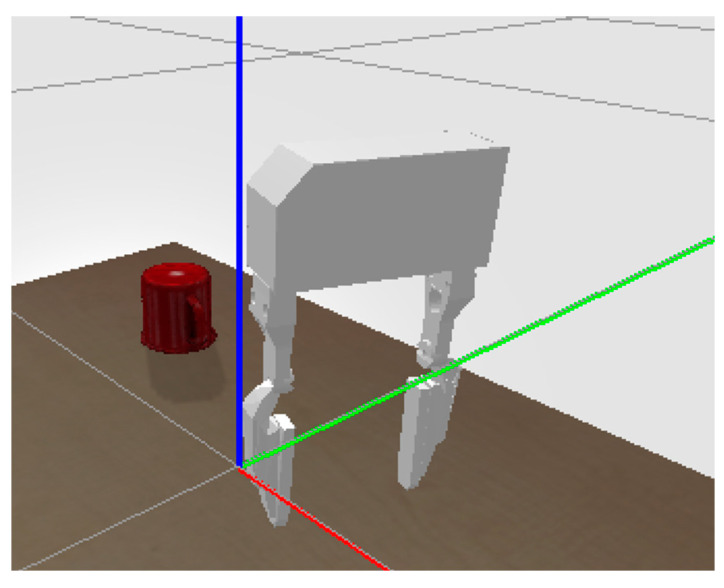
WSG50 model.

**Figure 4 bioengineering-11-00108-f004:**
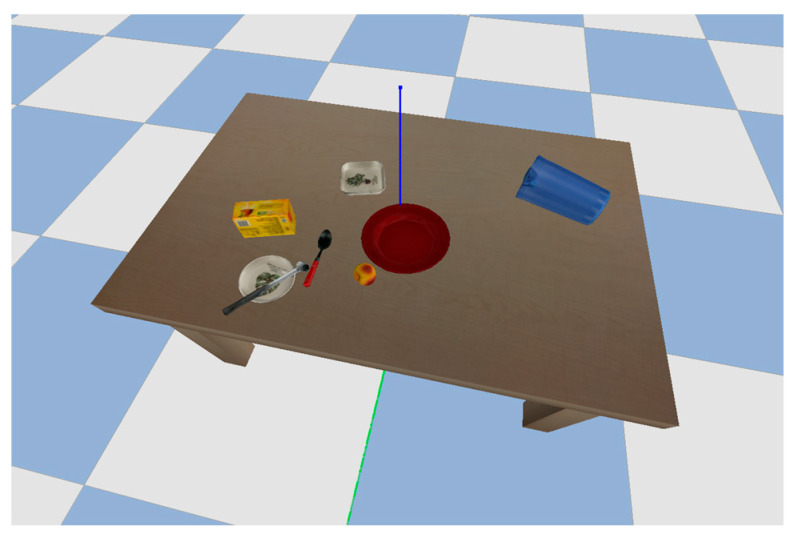
Some Pybullet URDF models.

**Figure 5 bioengineering-11-00108-f005:**
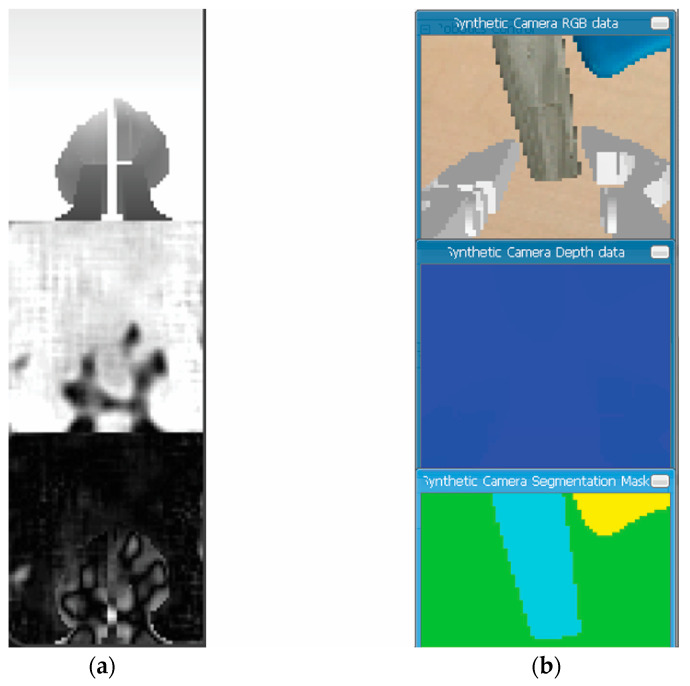
(**a**) Encoded and reconstructed depth image; (**b**) original depth image.

**Figure 6 bioengineering-11-00108-f006:**
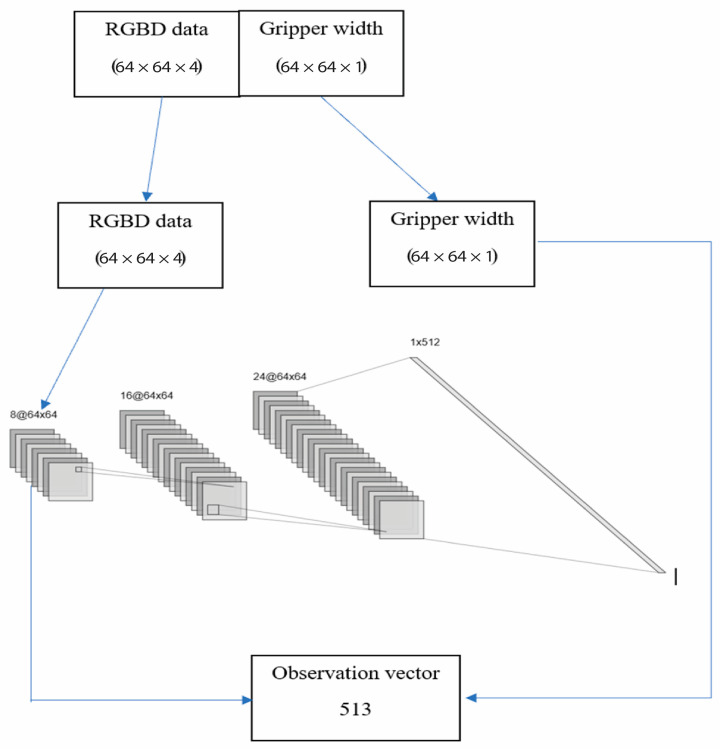
RGBD data into a convolutional neural network to form the observation vector for the Markov decision process.

**Figure 7 bioengineering-11-00108-f007:**
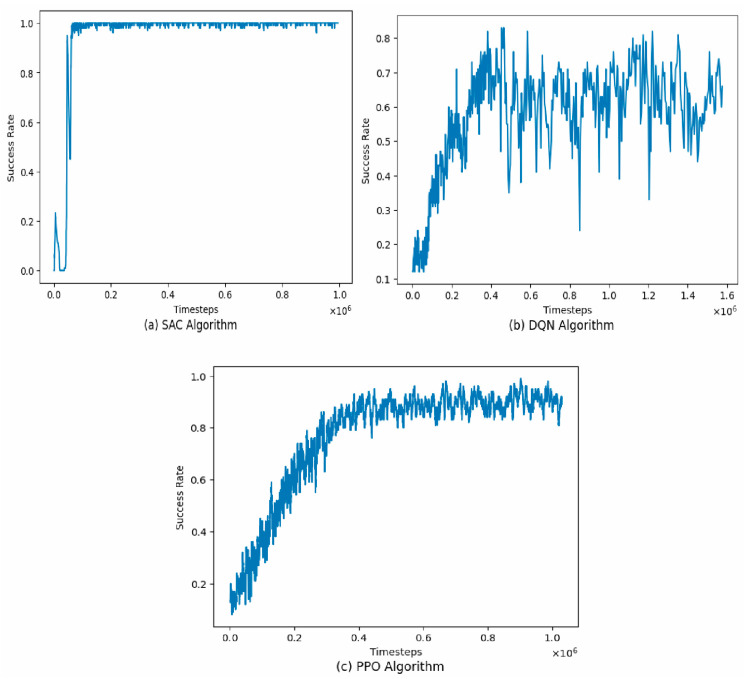
Comparative performance of SAC, DQN, and PPO algorithms: success rate by timesteps averaged across all objects: (**a**) SAC algorithm; (**b**) DQN algorithm; (**c**) PPO algorithm.

**Figure 8 bioengineering-11-00108-f008:**
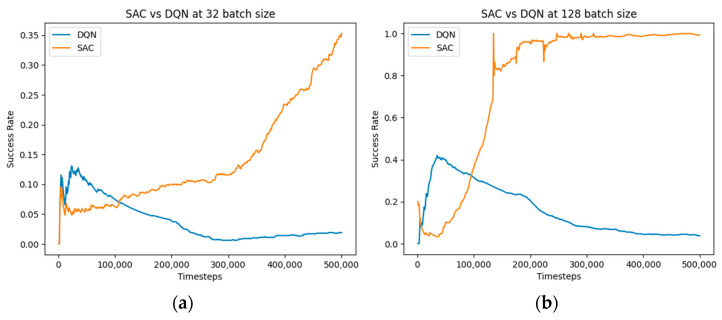
Comparative performance of SAC and PPO algorithms at batch sizes 128 and 32: (**a**) 32 batch size; (**b**) 128 batch size.

**Figure 9 bioengineering-11-00108-f009:**
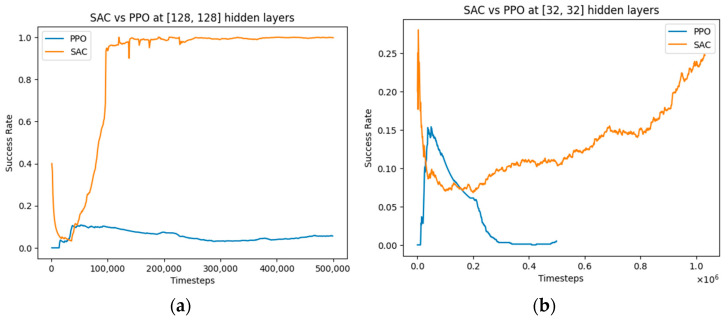
Comparative performance of SAC and PPO algorithms with a varied number of hidden layers: (**a**) [128, 128] hidden layers; (**b**) [32, 32] hidden layers.

**Figure 10 bioengineering-11-00108-f010:**
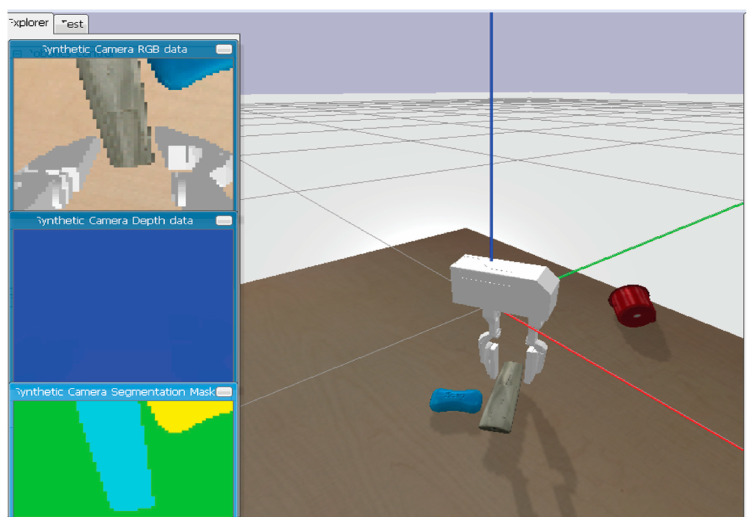
WSG50 gripper successfully grips the remote controller.

**Figure 11 bioengineering-11-00108-f011:**
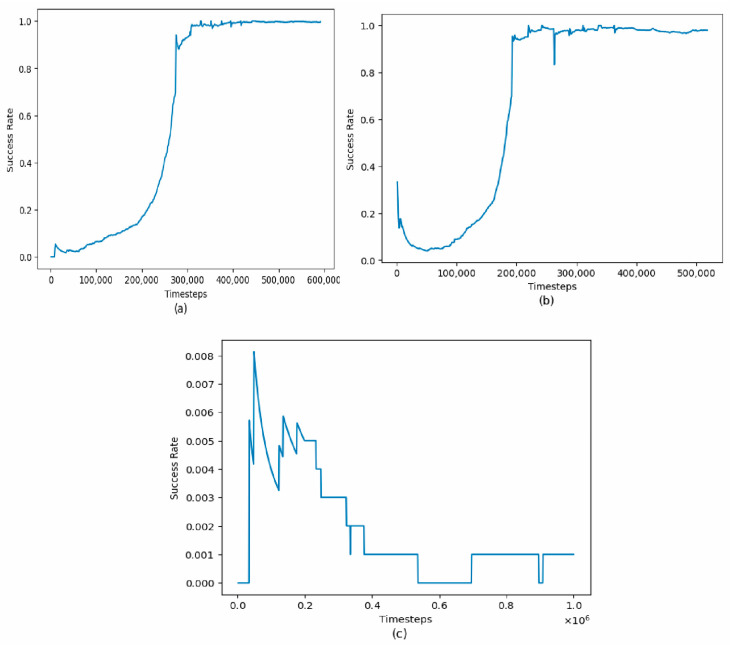
SAC success rate by timesteps in grasping: (**a**) a remote controller; (**b**) a mug; (**c**) a soap bar.

**Table 1 bioengineering-11-00108-t001:** Environment parameters.

	Environment
Action space	5
Observation	RGBD and Depth data
Reward	Shaped reward
Discount factor	0.99

**Table 2 bioengineering-11-00108-t002:** Detailed parameters of the custom-shaped reward function utilized to influence and optimize the gripper’s behavior in the environment.

	Non-Terminal State	Terminal State
Object grasped?	$r_{g}-r_{tp}$	$r_{t}-r_{tp}$
Not grasped?	$-r_{tp}$	@timeout $-r_{tp}$

$r_{t}\;$= terminal reward, $r_{g}$= grasping reward, $r_{tp}$ = time penalty.

**Table 3 bioengineering-11-00108-t003:** Mean success rate achieved by each algorithm (DQN, SAC, and PPO) in the experiment.

Summary	Result
Mean success rate for DQN	0.6021689086910577
Mean success rate for SAC	0.9903811107807406
Mean success rate for PPO	0.821488216618748

**Table 4 bioengineering-11-00108-t004:** MANOVA analysis results: assessing significant performance differences among DQN, SAC, and PPO algorithms.

	DF	Pillai	F-Value	Den DF	Pr (>F)
Algorithms	2	0.57982	2040.7	19,994	<2.2 × 10^−16^ ***
Residuals	9997				

*** indicates *p* < 0.001, denoting high significance.

**Table 5 bioengineering-11-00108-t005:** Tukey HSD post hoc results.

Group 1	Group 2	Mean Diff	P-adj	Lower	Upper	Reject
DQN	PPO	−0.2193	0.0	−0.2295	−0.2091	True
DQN	SAC	0.1689	0.0	0.1657	0.172	True
PPO	SAC	0.3882	0.0	0.3779	0.3985	True

**Table 6 bioengineering-11-00108-t006:** Friedman rank test.

	Result
*p*-value	0.0
Q-value	29,689.740691361738

**Table 7 bioengineering-11-00108-t007:** Comparative analysis of SAC, PPO, and DQN algorithms with varied number of hidden layers in the model architecture.

Algorithm	SAC	PPO	DQN
Convergence speed	Quickest to convergence	Slow convergence	Slow convergence
Hyperparameter sensitivity	Robust to hyperparameters, relatively easier to tune	Performance is heavily affected when deviating slightly from an optimal hyperparameter	Performance is also heavily affected
Training time	Takes the longest time to train	Lower training time than SAC	Requires the least amount of training time

**Table 8 bioengineering-11-00108-t008:** Results of the MANOVA test investigating the impact of different object types (a mug, a remote controller, a soap bar) on the success rates of the SAC algorithm.

	DF	Pillai	F-Value	Den DF	Pr (>F)
Algorithms	2	0.6888	552.9	4210	<2.2 × 10^−16^ ***
Residuals	2105				

*** indicates *p* < 0.001, denoting high significance.

## Data Availability

The data generated during the course of this study are available on GitHub at https://github.com/semtu/On-Automated-Object-Grasping-for-Intelligent-Prosthetic-Hands-Using-Machine-Learning (accessed on 9 December 2023).

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
