# Peer review of "On Automated Object Grasping for Intelligent Prosthetic Hands Using Machine Learning"

_bioengineering, 2024, doi:10.3390/bioengineering11020108_

Round 1
Reviewer 1 Report
Comments and Suggestions for Authors
As a reviewer of the paper titled "On Automated Object Grasping for Intelligent Prosthetic Hands Using Machine Learning," I recommend the authors for addressing a significant challenge in the field of prosthetics and attempting to enhance the usability and acceptance of electronic prosthetic hands using machine learning. However, Several aspects of the paper could be improved or require clarification:
· The paper should clearly delineate how its approach differs from or improves upon existing methods. The field of machine learning in prosthetics is rapidly evolving; thus, establishing the novelty of the proposed method is crucial.
· How the proposed methods performance compared to the state-of-the-art methods and how it goes beyond the state-of-the-art methods.
· In the introduction section, the research gaps are not clear.
· Figure 1 is blurry and needs to be improved.
· In the line 89 and 244, the (Baris, 2020) should be in sequence order with the rest of the references.
· While the choice of Soft Actor-Critic (SAC), Deep Q-Network (DQN), and Proximal Policy Optimization (PPO) is interesting, the paper should provide a more detailed rationale for selecting these specific algorithms. Furthermore, a comparison with other potential algorithms or approaches would strengthen the study.
· While a 99% success rate for SAC is impressive, the paper should clarify what criteria were used to define 'success' in the context of automated grasping.
· Comparative analysis against established benchmarks or baseline models would provide a clearer understanding of the algorithms' performance.
· More results show the real time demonstrations are required.
· The reinforcement learning should be better explained.
Comments on the Quality of English LanguageN/A
Author Response
Thank you for reviewing our manuscript, Please see the attachment.

Reviewer 2 Report
Comments and Suggestions for Authors
The authors proposed application of machine learning applications for automated object grasping. The electronic prosthetics, in general, requires training for users to gain motor control over prosthetic fingers, hindering their usability and acceptance. To address the same, the authors proposed an automated method that leverages computer vision-based techniques and machine learning algorithms. For the same, three reinforcement learning algorithms, namely Soft Actor-Critic (SAC), Deep Q-Network (DQN), and Proximal Policy Optimization (PPO), are employed to train agents for automated grasping tasks. In this regard, I have the following concerns:
1. The authors compared three algorithms, SAC, DQN, and PPO on the basis of MANOVA. The authors should also provide statistical analysis for the same.
2. For better comparison, the authors should provide non-parametric statistical test. In this regard, they should provide Friedman rank test listing the p-value and Q-value. The p-value and Q-value provide better comparison as far as non-parametric statistical tests are concerned.
3. Authors should also highlight if the proposed methodology may be applied in other areas other than Myo prosthetic.
4. The authors are also advised to improve the typographical and grammatical mistakes.
Comments on the Quality of English LanguageThe quality of the English language is acceptable.
Author Response

(The authors gave the same response as above.)

Reviewer 3 Report
Comments and Suggestions for Authors
This is a well-written paper regarding an original research on intellegent prosthetic hands using machine learning. The project is properly design and the conclusion is clear based on adequate result analysis.
Author Response
Thank you very much for taking the time to review our manuscript.
Reviewer 4 Report
Comments and Suggestions for Authors
Dear Authors,
Well done!
I am very happy to have reviewed this scientific paper.
I accept this paper as it is.
Author Response
Thank you very much for taking the time to review our manuscript. We have made some improvements in the results section which is highlighted in the re-submitted files
Reviewer 5 Report
Comments and Suggestions for Authors
The article "On Automated Object Grasping for Intelligent Prosthetic Hands Using Machine Learning" is applicable to be considered for publication in bioengineering journal. However the manuscript needs improvement in various section. Following are some issues that must be incorporated in the modified version of the article.
1. The abstract lacks background, problem statement and motivation behind carrying out the proposed study. In my opinion the abstract must reflects these to attract researchers and readers working in this domain.
2. Specifically, given a task to be performed with the prosthetic hand, how the human intention or desire to accomplish the task is mapped to the actuator on the prosthetic hand is a challenge . These lines may be rephrased to make it simpler one.
3. The aim and objectives of the presented study is missing in the research contribution statement (lines 76-80).
4. Figure 1 is not clear. The figure may be modified to enhance visibility.
5. The evaluation and the results of the proposed algorithm is missing in the conclusion section. The conclusion section must contains the research problem and the proposed method effectiveness.
6. The authors must check the overall article. Grammar mistakes and other typos must be corrected.
Comments on the Quality of English LanguageThe authors are requested to use simple and concise sentences
Author Response

(The authors gave the same response as above.)
